# Fabrication and Characterization of Electrospun Aligned Porous PAN/Graphene Composite Nanofibers

**DOI:** 10.3390/nano9121782

**Published:** 2019-12-15

**Authors:** Yanhua Song, Yi Wang, Lan Xu, Mingdi Wang

**Affiliations:** 1National Engineering Laboratory for Modern Silk, College of Textile and Engineering, Soochow University, 199 Ren-ai Road, Suzhou 215123, China; syanhua2015@163.com (Y.S.); 20185215036@stu.suda.edu.cn (Y.W.); 2School of Mechanical and Electric Engineering, Soochow University, Suzhou 215123, China; wangmingdi@suda.edu.cn

**Keywords:** graphene, aligned nanofibers, porous nanocomposites, electrospinning

## Abstract

A modified parallel electrode method (MPEM), conducted by placing a positively charged ring between the needle and the paralleled electrode collector, was presented to fabricate aligned polyacrylonitrile/graphene (PAN/Gr) composite nanofibers (CNFs) with nanopores in an electrospinning progress. Two kinds of solvents and one kind of nanoparticle were used to generate pores on composite nanofibers. The spinning parameters, such as the concentration of solute and solvent, spinning voltage and spinning distance were discussed, and the optimal parameters were determined. Characterizations of the aligned CNFs with nanopores were investigated by scanning electron microscopy (SEM), fourier transform infrared (FTIR) spectroscopy, X-ray diffraction (XRD) analysis, transmission electron microscopy (TEM), high-resistance meter, and other methods. The results showed that graphene (Gr) nanoparticles were successfully introduced into aligned CNFs with nanopores and almost aligned along the axis of the CNFs. The MPEM method could make hydrophobic materials more hydrophobic, and improve the alignment degree and conductive properties of electrospun-aligned CNFs with nanopores. Moreover, the carbonized CNFs with nanopores, used as an electrode material, had a smaller charge-transfer resistance, suggesting potential application in electrochemical areas and electron devices.

## 1. Introduction

Supercapacitors are promising energy storage devices with high power density and long cycle life, and they are of considerable interest for applications in hybrid electrical vehicles, pulsed laser system and so on [1,2,3]. Among various electrode candidates for supercapacitors, porous carbons are mainly investigated due to their advantages, such as well-developed nanostructure, excellent physicochemical stability, good conductivity and cost-effectiveness [4,5]. However, electrochemical behaviors of supercapacitors are influenced by the nature of carbon materials, such as exposed surface area, pore size distribution, functional groups on carbon surfaces and electronic conductivity [6,7]. Therefore, tailoring the porous structure and surface chemistry of activated carbons is of great importance to improve the electrochemical performance of supercapacitors [8,9,10].

Electrospinning (ES) has been proven to be an efficient and controllable method for the fabrication of nanofibers with nanopores [11,12]. Electrospinning has the advantages of a simple process, convenient operation, applicable to various materials, and its products have the structural characteristics of controllable pore diameter and specific surface area [12,13,14]. The porous effect of the electrode material can not only increase the specific surface area of the material, but also provide a channel for the transfer and transport of ions in the supercapacitor, resulting in larger capacity of the supercapacitor [6,7,8,9,10]. Therefore, recent researches showed electrospun nanofibers with pores could be fabricated simply by adjusting electrospinning parameters to improve the porous effect of the nanofibers [11,15,16] for the capacitor. However, the electrospun porous nanofibers usually show randomly oriented structures and low alignment, which cld result in weak conductivity and low molecular orientation, limiting their practical uses [17]. The aligned structure of porous nanofibers can make the electrode material have aligned nanostructures, which would promote the ion transfer, save transfer time, improve the transport efficiency and enhance the conductivity. Some literatures reported that the aligned mesoporous carbons have superior capacitive behavior, power output and high-frequency performance in supercapacitors due to their mesopore network structure [18,19,20].

In addition, the properties of the nanofibers prepared using a single polymer are relatively poor and cannot meet the requirements of practical applications. By contrast, polymeric nanocomposites with nanoparticles usually exhibit terrific properties, and have been widely used in recent years, for purposes ranging from biomedical [21,22,23,24], microwave absorption [25,26], electrochemical [27,28], and optical materials [29]. Presently, it is a hot subject to broaden the application field of nanocomposites with good electrical conductivity in science and engineering. Graphene (Gr), with a two-dimensional (2D) structure, one-atom thickness and honeycomb-like sheet of sp^2^-hybridised carbon atoms with a conjugated system of overlapping π electrons [30,31], has sparked massive interests in many research teams worldwide and has revolutionized the scientific frontier in nanoscience and condensed matter physics because of its extraordinarily electrical, physical, and chemical properties [32]. It has been an excellent candidate for electronic devices [33], gas sensors [34] and electrochemical devices. Electrospun porous composite nanofibers (CNFs) have been widely used as electrode, supercapacitor and catalyst, etc. in the electrochemical field [35,36]. Both polyacrylonitrile (PAN) and Gr are favorable electrical materials that can be applied as carbon-based materials to fabricate electrochemical devices [37,38,39]. Porous carbon-based composites have features of relatively light quality, outstanding electricity and full absorption band. In addition, PAN/Gr CNFs are also widely used in other important areas such as water purification [40,41] and advanced carbon yarns [42]. However, these PAN/Gr CNFs all do not have aligned and nanoporous structure.

In our previous works [43,44,45], a modified parallel electrode method (MPEM) was presented to fabricate highly aligned electrospun nanofibers for a long spinning time, where a positively charged ring was placed between the needle and the two paralleled metal electrode receivers. The resultant force produced by the ring would increase the kinetic energy of the moving jet, accelerate the downward movement of the jet, and shrink the radius of the whipping circle. As a result, the stability condition and the nanofiber alignment were improved and the diameter became much smaller. That meant the MPEM could decrease the nanofiber diameter, enhance the diameter distribution, and improve the nanofiber alignment. Therefore, in this study, aligned nanoporous PAN/Gr CNFs with different Gr concentrations were prepared by the MPEM, and the solution system was made up of PAN, Gr, N, N-dimethylformamide (DMF) and water. The effects of electrospinning parameters, such as spinning voltage, spinning distance, the concentration of PAN and the water contents on the electrospun CNFs were studied, and the optimum parameters were determined. Morphologies, crystalline structures and conductive properties of the aligned CNFs with nanopores were investigated by scanning electron microscopy (SEM), Fourier transform infrared (FTIR) spectroscopy, X-ray diffraction (XRD), transmission electron microscopy (TEM), contact angle (CA) measurements and high-resistance meter. The results showed that the obtained PAN/Gr CNFs using MPEM exhibited high orientation, porosity and hydrophobicity when the optimum spinning parameters were adopted. The carbonized CNFs with nanopores, used as an electrode material, had a smaller charge-transfer resistance, which improved their electrochemical performance when used as the capacitor electrodes.

## 2. Experimental

### 2.1. Materials

PAN powder (*M*_W_ = 150,000), was supplied by Beijing Lark Branch Co. Ltd. (Beijing, China). N, N-dimethylformamide (DMF) (Analytical Reagent) was purchased from Shanghai Chemical Reagent Co. Ltd. (Shanghai, China). Deionized water (H_2_O) and Gr nanoplatelets (thickness: 6–8 nm, width: 5μm) were purchased from Shanghai Aladdin Biochemical Technology Co. Ltd. (Shanghai, China). Syringes with capacity of 10 mL and needles with diameter of 0.7 mm and length of 32 mm were purchased from Shanghai Misawa Medical Industry Co. Ltd. (Shanghai, China). All materials were used without any further purification.

### 2.2. Preparation of Spinning Solution

All the solution concentration ratios were weight to weight (*w*/*w*). Gr nanoplatelets with different concentrations of 0 wt%, 0.5 wt%, 1.5 wt% and 2.5 wt% were dispersed in a DMF/H_2_O solvent system, respectively, using an ultrasonic cleaner (SL-5200DT, Nanjing Shunliu Instrument Co. Ltd., Nanjing, China) for 0.5 h at 25 ± 2 °C (room temperature). The concentrations of H_2_O were 0 wt%, 2 wt%, 5 wt% and 8 wt%. Then the spinning solutions were prepared by dissolving PAN with different concentrations of 8 wt%, 10 wt% and 12 wt% respectively in Gr/DMF or Gr/H_2_O/DMF solution under magnetic stirring for 24 h at room temperature (25 ± 2 °C) until it became homogeneous. All the concentrations were related to the spinning solution.

### 2.3. Fabrication of Aligned Nanoporous PAN/Gr CNFs

Aligned and random nanoporous PAN/Gr CNFs with the Gr concentrations ranging from 0 to 2.5 wt% were prepared directly by MPEM and ES at room temperature and relative humidity of 50 ± 5%. The spinning apparatus is shown in Figure 1 [45]. The whole devices included two direct current high-voltage power generators (0–30 kV, DW-P303-1ACF0, Tianjin DongWen high-voltage power generator Co., Ltd., Tianjin, China), a syringe, a needle, a parallel electrode collector and a flow pump (Longerpump Co., Ltd., Baoding, China). The effects of different spinning voltages, spinning distances (needle to receivers), distances of two paralleled electrodes and the ring voltages on electrospun CNFs were investigated. The needle and copper ring were applied different positive voltages and the paralleled auxiliary electrodes were connected to negative pole. The ring was 21 cm in diameter and the flow rate was 0.8 mL/h. The applied spinning voltage varied from 12 to 24 KV. The applied ring voltage varied from 3 to 7 KV. The distance of two paralleled electrodes varied from 1 to 5 cm. The spinning distance varied from 12 to 24 cm. Then the electrospun nanofibers were obtained on a collector, and a large number of nanofibers form nanofiber membranes.

### 2.4. Measurements and Characterizations

#### 2.4.1. Property Characterizations of Spinning Solutions

The viscosity and conductivity of spinning solutions were measured by a viscometer (SNB-1, Shanghai Fangrui Instrument Co. Ltd., Shanghai, China) and a conductivity meter (DDS-307, Shanghai instrument & electric Scientific Instrument Co., Ltd., Shanghai, China), respectively. The measurement was repeated three times.

#### 2.4.2. Morphology and Structure Characterizations of Electrospun CNFs

The morphologies of PAN/Gr CNFs were examined by a scanning electron microscopy (SEM, Hitachi S-4800, Tokyo, Japan) at an acceleration voltage of 3 kV. The samples were dried before being sputter-coated with gold. The fibrous diameters of PAN/Gr CNFs were calculated by ImageJ software (National Institute of Mental Health, Bethesda, MD, USA). The alignment degree of CNFs was measured by ImageJ software according to the offset angle (*θ*) between the long axis of the nanofiber and its expected direction (the vectors of parallel electric field). The ratio of the number of these nanofibers, whose θ between −9° and 9° was compared to the total number of all nanofibers.

A transmission electron microscopy (TEM, FEI, Tecnai G-20, Hillsboro, OR, USA) was used to characterize with the morphologies’ structures and the distribution of Gr in the CNFs. TEM worked at a high voltage of 200 kV, a dark current of 10.57 μΑ, and an emission current of 64 μΑ.

The structure of CNFs and the interactions of polymer with Gr were investigated through FTIR spectroscopy (Nicolet5700, Thermo Nicolet Company, Madison, WI, USA). For each measurement, each spectrum was obtained by the performance of 32 scans with the wavenumber ranging from 400 to 4000 cm^−1^ and a resolution of 4 cm^−1^.

The crystalline structure of PAN, Gr and CNFs were elucidated by X-ray diffraction (XRD, Philips X’Pert-Pro MPD, PANalytical, Almelo & Eindhoven, Almelo, The Netherlands) with a 3 kW ceramic tube as the X-ray source (Cu-Kα) and an X’Celerator detector. The diffraction angle 2θ range was 5°–60°, and the diffraction patterns were collected at a scanning rate of 4°/min.

#### 2.4.3. Porosity, Wetting Property and Conductivity Characterizations of Electrospun CNFs

The pore size distributions of aligned and random porous PAN/Gr CNF membranes were investigated by a capillary flow porometry (Porometer 3G, Quan-tachrome Instruments, Boynton Beach, FL, USA), which applied the technique of expelling Porofil, a wetting liquid, through pores in the membranes. All CNF samples were circular membranes with a diameter of 25 mm and thickness of 10 μm.

The wetting properties were characterized by testing water static contact angle (CA) of porous PAN/Gr CNFs obtained by ES and MPEM using a Krüss DSA 100 apparatus (Krüss Company, Brandenburg, Germany). The volume of droplets used for static CA was 6 μL. The average water CAs were calculated by measuring the same sample at least in five different positions [46].

To determine the electrical conductivity of the aligned and random CNFs, the surface resistance was measured by the high-resistance meter (ZC36, Shanghai, China) at room temperature and ambient condition.

#### 2.4.4. The Charge-Transfer Resistances of the Carbonized CNFs

The electrospun CNFs were placed in a muffle furnace (GZ2.5-10TP, Shanghai Gaozhi Precision Instrument Co., Ltd., Shanghai, China) at 280 °C for 2 h in an air atmosphere to be oxidized and stabilized. The oxide-stabilized CNFs were carbonized at 1100 °C in a horizontal furnace (OTF-1200X-II, Hefei Kejing Material Technology Co., Ltd., Hefei, China) under a flow of nitrogen at a heating rate of 5 °C/min to obtain the carbonized CNFs. Finally, an autolab electrochemical station (PGSTAT302N, Metrohm, Herisau, Switzerland) was used to measure the charge-transfer resistances of the carbonized CNFs.

## 3. Results and Discussion

### 3.1. Property Characterizations of Spinning Solutions

The viscosity and conductivity of the spinning solution have significant influences on the spinnability and spinning effect of the solution [44]. As shown in Figure 2, when the concentration of water was 2 wt %, both the viscosity and conductivity of solutions were enhanced as the concentrations of PAN and Gr in the solutions increased. Gr were dispersed evenly in the PAN solution, obtaining a stable suspension. However, with the increase of the Gr concentration, the dispersion uniformity of Gr would degrade, and aggregation of Gr would occur, leading to a gradual increase in the viscosity of the solution.

Compared with the increase of solution viscosity, the increase of solution conductivity was minimal. Therefore, the change of solution conductivity would have little effect on the experimental results in the subsequent experiments. However, the viscosity played an essential role in determining the diameter of electrospun nanofibers.

### 3.2. Morphological Characterization of Aligned and Nanoporous PAN/Gr CNFs (SEM)

#### 3.2.1. Effects of the Concentrations of Solute and Solvent on the Electrospun CNFs

According to [35] and our previous work [43,44], the effects of the concentrations of solute and solvent on the electrospun CNFs were investigated when the spinning voltage was 18 kV, the ring voltage was 5 kV, the distance of two electrodes was 5 cm and the spinning distance was 18 cm.

First, the concentrations of Gr at 0.5 wt% and PAN at 12 wt% were selected to investigate the effects of different H_2_O concentrations on the nanopore formation in the electrospun PAN/Gr CNFs. As exhibited in Figure 3, there appeared visible nanopores on the CNFs surface after the addition of H_2_O. The reason for this is that DMF is an organic solvent and easy to volatilize rapidly in the spinning process, yet the volatility of H_2_O is lower in the spinning process. That means the volatilization time of H_2_O and DMF is inconsistent, resulting in the pore formation on the CNFs surface. With the H_2_O concentration increased from 2 to 5 wt%, the nanofiber diameter became thicker and lumps were found. With the H_2_O concentration further increased, the spinning process became more and more difficult, and even when the H_2_O concentration was higher than 8 wt%, the solution could not be spun to nanofibers. The reason was that PAN couldn’t be dissolved by H_2_O, and with the H_2_O concentration increased, the solution became nonhomogeneous and a beaded structure appeared. Nanofibers with nanopores have a relatively higher surface area compared to nanofibers without nanopores, leading to broader application. Therefore, 2 wt% was selected as the best H_2_O concentration in subsequent study.

The aligned and nanoporous electrospun PAN/Gr CNFs with a Gr concentration of 0.5 wt% were prepared by the addition of 2 wt% H_2_O, and the effects of PAN concentration on the CNFs were investigated. From Figure 4, it was evident that with the increase of PAN concentration from 8 to 12 wt%, the average diameter also increased from 368.8 to 579.9 nm, but the beads on the CNFs gradually decreased until they disappeared. Additionally, the alignment degree of nanofibers was improved from 47.1 to 76.0%. That’s because, with the increase of the viscosity, the polymer chains resist electric field-stretching, and the bending instability of jets can be suppressed. As a result, both the diameter and alignment degree of composite nanofibers increased [44,45]. Thus the PAN concentration of 12 wt % was chosen as the future MPEM parameter.

Figure 5 showed that with the Gr concentration increased, more and more beads could be found, and when the Gr concentration was 2.5 wt%, there were many lumps in the CNFs. This phenomenon could be related to the aggregation of Gr, and it was essential to get a well-dispersed Gr solution [47]. Therefore, the Gr concentration of 0.5 wt% was selected as the subsequent MPEM parameter.

#### 3.2.2. Effects of the Spinning Process Parameters on the Electrospun CNFs

According to the above results, when the PAN concentration was 12 wt%, the Gr concentration was 0.5 wt % and the H_2_O concentration was 2 wt%, the effects of the spinning process parameters, such as the spinning voltage, the ring voltage, the distance of two paralleled electrodes and the spinning distance on the electrospun CNFs, were studied by SEM. Figure 6 shows that with the increase of the spinning voltage, the average diameter of electrospun CNFs decreased, and the alignment degree firstly increased and then decreased. When the spinning voltage was 12 kV, the spinning process was relatively slow, the collected CNFs were widely distributed, and the alignment degree of CNFs was 73.3%. When the spinning voltage was 18 kV, a large number of CNFs could be collected, the fiber diameter distribution was more uniform, and the alignment degree was significantly increased to 90.5%. When the spinning voltage increased to 24 kV, the jet velocity evidently accelerated, the CNFs appeared as nodules, and the alignment degree decreased to 72.2%. That meant that when the applied voltage was too low or too high, it would affect the stability of the jet and make the uniformity of diameter distribution and alignment degree of CNFs obtained worse. Therefore, when the applied voltage was 18 kV, the diameter distribution of CNFs had excellent uniformity, and their alignment degree could reach 90.5%.

The alignment degrees of CNFs with ring voltages of 3 kV, 5 kV and 7 kV were 60.6, 90.5 and 53.3%, respectively. In Figure 7, it was apparent that with the increase of the ring voltage, the uniformity of diameter distribution and the alignment degree of CNFs first increased and then decreased. This was because when the ring voltage was too small or large, the corresponding repulsion of the charged jet was too small or large, which could result in unstable motion of the jet. As a consequence, the uniformity of diameter distribution and the alignment degree of CNFs would become worse. According to the results, 5 kV was the optimal ring voltage.

From Figure 8, it could be directly seen that the alignment degree of CNFs with an electrode distance of 5 cm was highest (82.1%). It was possible that when the distance of two metal electrodes was narrower than 5 cm, the repulsive force of the charged jet was too large, resulting in a low alignment degree.

From Figure 9, when the spinning distance was 12 cm, the collected CNFs were distributed widely, the alignment degree of CNFs was 54.2%, and there were no nanopores on the surface of CNFs. When the spinning distance increased to 18 cm, the number of CNFs obtained increased significantly, and nanopores appeared on their surface. The reason was that when the spinning distance increased, the jet movement time increased, and the solvent volatilized sufficiently, resulting in the appearance of nanopores and the increase of diameter. At this time, the alignment degree of CNFs was the highest at 89.5%. When the spinning distance reached 24 cm, the fiber distribution was scattered and sparse, and the alignment degree decreased to 73.2%. This was because as the spinning distance increased, there was not enough electric field force to draft the jet due to the decrease of the electric field strength, leading to the less CNFs received and the inferior alignment degree of CNFs. Therefore, the spinning distance of 18 cm was selected as the optimal spinning distance in the MPEM process.

According to the above SEM results, the PAN concentration of 12 wt%, Gr concentration of 0.5 wt %, H_2_O concentration of 2 wt%, spinning voltage of 18 kV, ring voltage of 5 kV, distance of two electrodes of 5 cm, and spinning distance of 18 cm were chosen as the optimum MPEM parameters. For comparison, random CNFs were prepared by ES using the same parameters.

### 3.3. TEM Analysis

The TEM method was utilized to determine the distribution of Gr in the aligned PAN/Gr CNFs with nanopores. From Figure 10, it could be seen that the dark shaded parts of the CNFs were the Gr sheets with length of 5 μm. That meant Gr was introduced into the CNFs successfully and almost aligned along the CNFs axis.

### 3.4. FTIR Spectra Analysis

The FTIR spectrum was used to investigate the intermolecular interactions between PAN and Gr. As shown in Figure 11, minor changes in the spectra of the pure PAN nanofiber upon the addition of Gr were found. One of these subtle modifications was observed in the 3300–3500 cm^−1^ region, indicating that the π electrons present in Gr interact with the hydrogen (free and bonded) attached to the nitrogen in the urethane bond, thus changing the shape of the band, or because Gr used in the experiments might not be restored fully and have hydroxyl on it.

In the spectrum of PAN, the bands at 1239, 1378 and 1452 cm^−1^ belonged to the C–H bending of PAN, the band at 2243 cm^−1^ was attributed to CN stretching vibration, and the peak at 1668 cm^−1^ was due to C=C stretching vibration of PAN. In the Gr spectra, the peak at 1378 cm^−1^ corresponded to C–O–C, which confirmed that Gr used in the experiments wasn’t restored completely. The graphene used in the experiments belonged to redox graphene, which were prepared by the reduction of graphite oxide. In the fabrication process, the graphene oxide might not be completely reduced, resulting in the presence of the oxygen-containing group of C–O–C in the Gr.

### 3.5. XRD Spectra

From the spectra results in Figure 12,the sharp peak observed at 2*θ* = 17° could be assigned as (200) crystal planes of PAN [48]. The peaks at 2*θ* = 26.5°, 44.5°, 54.5° indicated the typical signal of Gr or graphite structures [49,50]. This peak was associated with the (002) diffraction of the hexagonal graphite structure in the carbon materials. The XRD spectra of PAN/Gr had both typical peaks of PAN and Gr and there were no new crystalline peaks in the PAN/Gr CNFs. Gr and PAN still retained their crystalline structures. The weak peak of the graphene structure at 44.5° did not appear, probably because of the high ratio of PAN composite with respect to Gr.

### 3.6. Porosity, Wetting, and Conductivity Properties

The porosity of CNF membranes were measured by a capillary flow porometry, in which the pores of the CNF membrane refered to the spacings between the CNFs and the spacings were assumed to be spherical pores. Table 1 and Figure 13 show that with the addition of Gr and the appearance of nanoporous structure on the surface of the CNFs, the pore size of the CNF membrane decreased and the corresponding pore number increased. At the same time, the MPEM method endowed the aligned CNF membrane with smaller pore size but greater pore number.

From Table 2, it can be concluded that the addition of Gr and the emergence of nanoporous structure on the surface of the CNFs made the contact angle of the CNF membrane larger and the hydrophobicity stronger. The main reason was that the addition of Gr and the appeared nanoporous structure on the CNF surface resulted in a rougher surface of the CNF membrane. As a result, the contact areas between H_2_O and the surface of the CNF membrane became smaller. According to Cassie–Baxter theories [43], the smaller contact area between H_2_O and materials leads to a larger contact angle and stronger hydrophobicity of the materials. Therefore, compared with the PAN/Gr CNF membrane without nanopores, the contact area of the PAN/Gr CNF membrane with nanopores was smaller according to Table 1, resulting in its larger contact angle and stronger hydrophobicity. In addition, it could be seen from Table 2 that the contact angle of random CNF membrane was smaller than that of the aligned CNF membrane due to the bigger pore size of the random CNF membrane as shown in Table 1. The bigger pore size meant a longer distance of the adjacent nanofibers. According to geometrical potential in capillaries, a longer distance of the adjacent nanofibers predicted a weaker repelling force [43]. As a result, its water CA became smaller. Therefore, the MPEM method could provide hydrophobic materials more hydrophobicity [43].

Then, the surface resistances of both random and aligned PAN/Gr CNF membranes were investigated to characterize electrical conductivity, as illustrated in Table 3. It was apparent that the surface resistance of aligned CNF membrane was lower than that of random CNF membrane, which demonstrated that the alignment degree of nanofibers had a profound effect on the electrical conductivity. However, the resistance of the CNF membrane with nanopores was higher than that of the CNF membrane without nanopores under the same conditions, which might be due to the nonconductive air filling the nanopores on the CNF surface, resulting in reduced conductivity.

### 3.7. The Charge-Transfer Resistances of Carbonized CNFs

The lower the charge-transfer resistance of the electrode material, the higher the transfer efficiency of ions in electrolyte. To further explore the effects of nanoporous and aligned structure of electrospun CNFs on the electrochemical performances as the electrode materials, the charge-transfer resistances of the carbonized CNFs as electrode materials were determined by measuring the intrinsic electrical resistance of the material. As shown in Table 4, the internal resistance of the carbonized aligned CNFs was obviously lower than that of the carbonized random CNFs, which exhibited that the alignment degree of nanofibers had a great influence on the internal resistance of the electrode material. In addition, the carbonized aligned CNFs with nanopores had the lowest internal resistance, which indicated that the nanopores could reduce the charge-transfer resistance, leading to better electrochemical performances as the electrode materials.

## 4. Conclusions

In this study, aligned PAN/Gr CNFs with nanopores were successfully fabricated by the MPEM method, in which a conductive copper ring was added between the needle tip and two paralleled metal electrodes collector. In the MPEM process, the effects of spinning parameters such as the concentrations of PAN, Gr, and H_2_O, spinning voltage, ring voltage, distance of two paralleled electrodes, and spinning distance on the properties of electrospun CNFs were investigated by SEM, FTIR, XRD, TEM and other instruments.

SEM pictures indicated that evident pores appeared on the CNFs surface with the addition of H_2_O, and the aligned CNFs with nanopores were prepared by MPEM using optimal spinning parameters, which were a PAN concentration of 12 wt%, Gr concentration of 0.5 wt%, H_2_O concentration of 2 wt%, spinning voltage of 18 kV, ring voltage of 5 kV, distance of two paralleled electrodes of 5 cm and spinning distance of 18 cm. FTIR, XRD and TEM data displayed Gr was introduced into the aligned CNFs with nanopores successfully and almost aligned along the axis of the CNFs. Moreover, the measurement results of pore size and contact angle of CNF membranes exhibited the MPEM could make hydrophobic materials more hydrophobic.

In addition, the surface resistance results illustrated that the alignment of CNFs could improve electrical conductivity. Further, the carbonized aligned CNFs with nanopores had the lowest internal resistance, which indicated that the nanopores could enhance electrochemical performances as electrode materials, resulting in high surface area and energy storage property.

## Figures and Tables

**Figure 1 nanomaterials-09-01782-f001:**
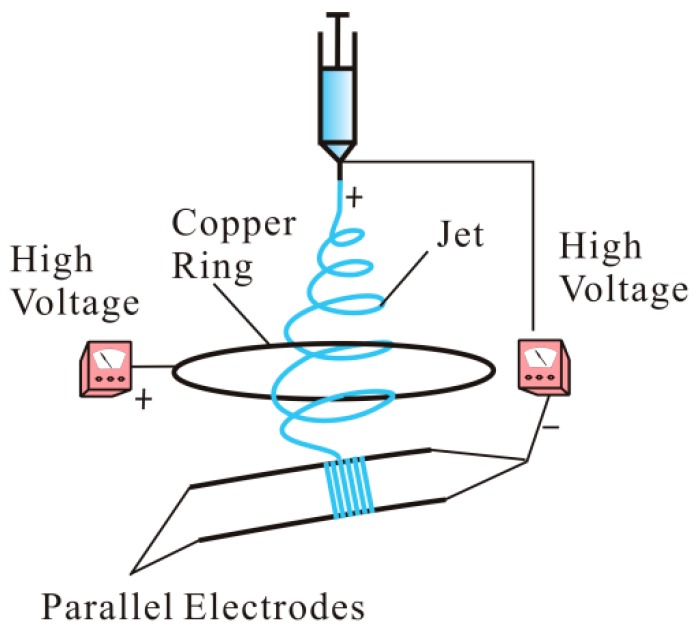
Schematic of the modified parallel electrode method (MPEM) apparatus.

**Figure 2 nanomaterials-09-01782-f002:**
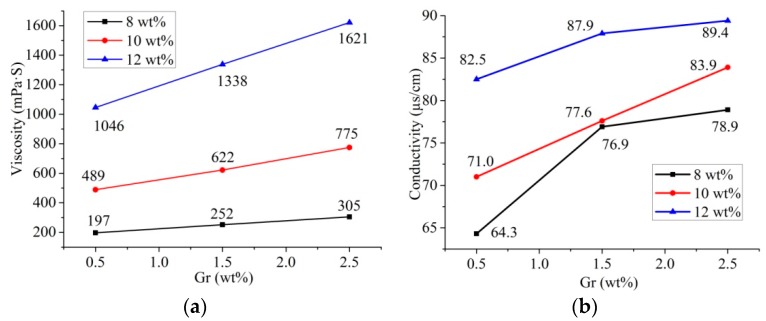
Viscosity (**a**) and conductivity (**b**) of spinning solutions with different concentrations of polyacrylonitrile (PAN) and graphene (Gr).

**Figure 3 nanomaterials-09-01782-f003:**
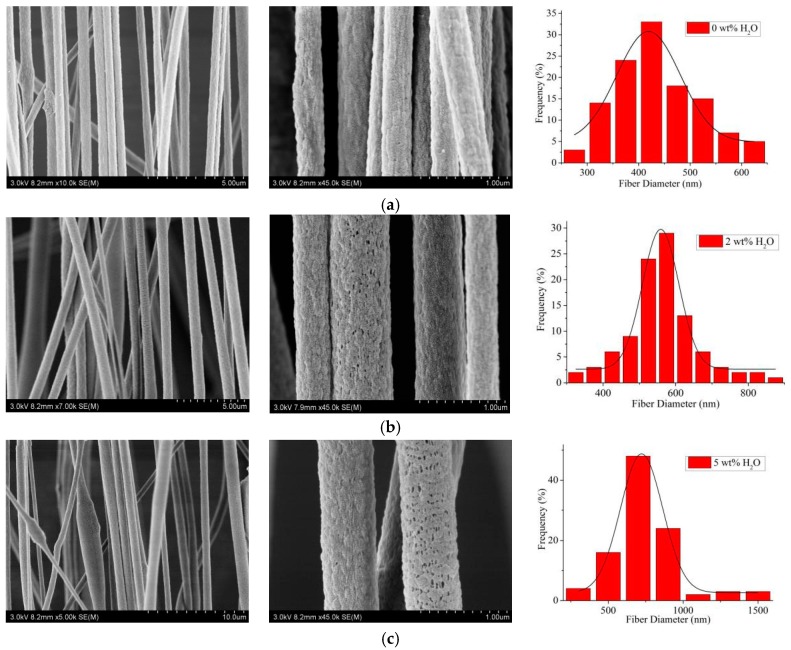
SEM pictures and the according diameter distribution (rightmost) of aligned porous CNFs with different concentrations of H_2_O by MPEM (Average diameter: (**a**) 0 wt% H_2_O, 406.2 ± 11.1 nm; (**b**) 2 wt% H_2_O, 561.730 ± 18.9 nm; (**c**) 5 wt% H_2_O, 754.64 ± 43.0 nm).

**Figure 4 nanomaterials-09-01782-f004:**
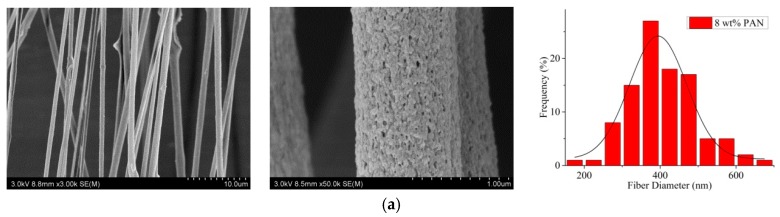
SEM pictures and the according diameter distribution (rightmost) of aligned porous CNFs with different PAN concentrations by MPEM (Average diameter: (**a**) 8 wt% PAN, 368.8 ± 17.3 nm; (**b**) 10 wt% PAN 406.4 ± 11.1 nm; (**c**) 12 wt% PAN 579.9 ± 26.6 nm).

**Figure 5 nanomaterials-09-01782-f005:**
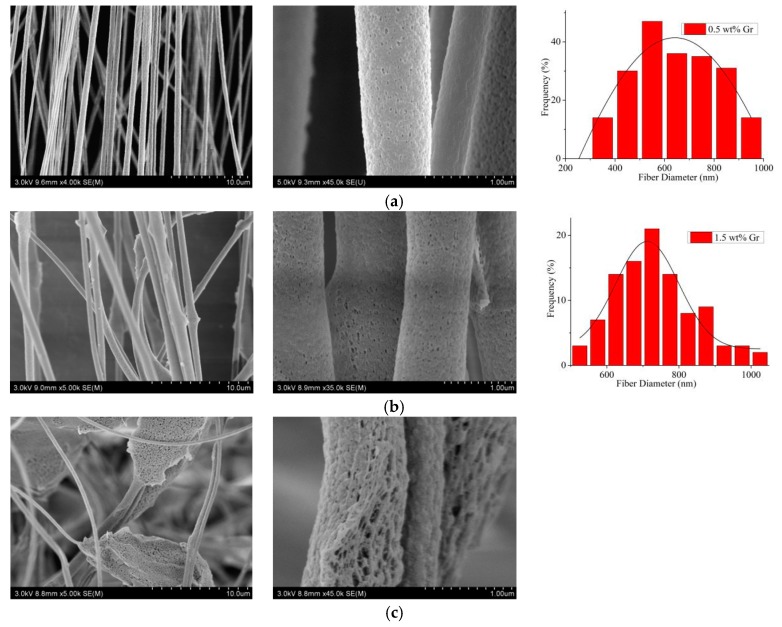
SEM pictures and the according fiber diameter distribution (rightmost) of aligned porous CNFs with different concentrations of Gr by MPEM (Average diameter: (**a**) 0.5 wt% Gr, 579.9 ± 17.3 nm; (**b**) 1.5 wt% Gr, 733.1 ± 21.6 nm; (**c**) 1.5 wt% Gr).

**Figure 6 nanomaterials-09-01782-f006:**
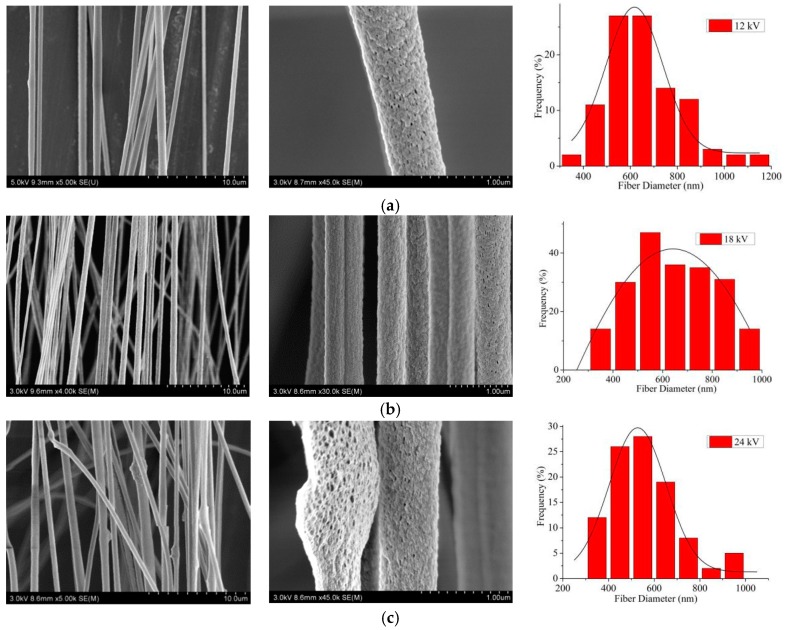
SEM pictures and the according diameter distribution (rightmost) of aligned porous CNFs with different spinning voltages by MPEM (Average diameter: (**a**) 12 kV, 665.1 ± 31.0 nm; (**b**) 18 kV,579.9 ± 26.6 nm; (**c**) 24 kV, 559.1 ± 28.4 nm).

**Figure 7 nanomaterials-09-01782-f007:**
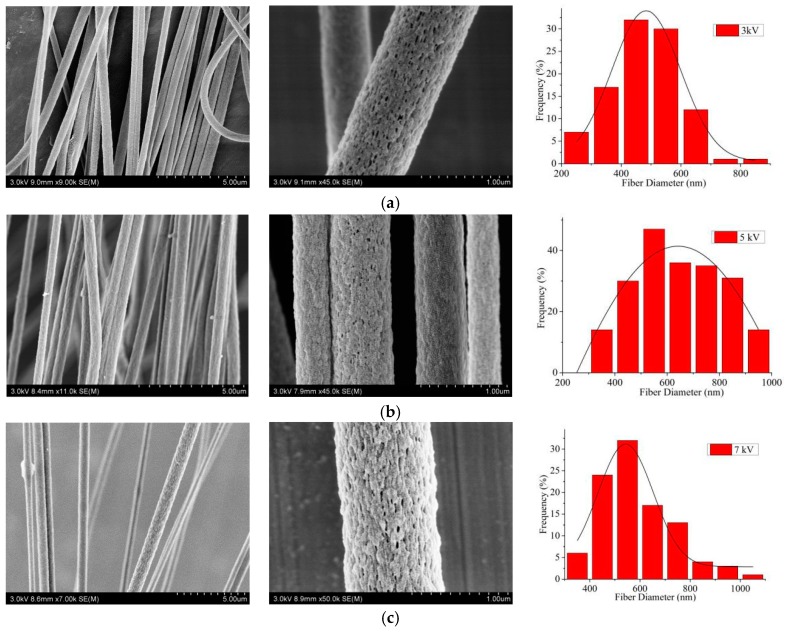
SEM pictures and the according diameter distribution (rightmost) of aligned porous CNFs with different ring voltages by MPEM (Average diameter: (**a**) 3 kV, 486.2 ± 22.4 nm; (**b**) 5 kV, 579.9 ± 26.6 nm; (**c**) 7 kV, 592.1 ± 28.7 nm).

**Figure 8 nanomaterials-09-01782-f008:**
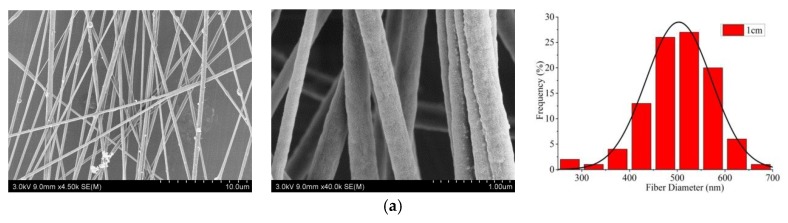
SEM pictures and the according diameter distribution (rightmost) of aligned porous CNFs with different distances of two metal electrodes by MPEM (Average diameter: (**a**) 1 cm, 503.3 ± 13.4 nm; (**b**) 3 cm, 504.5 ± 17.8 nm; (**c**) 5 cm, 510.3 ± 18.1 nm).

**Figure 9 nanomaterials-09-01782-f009:**
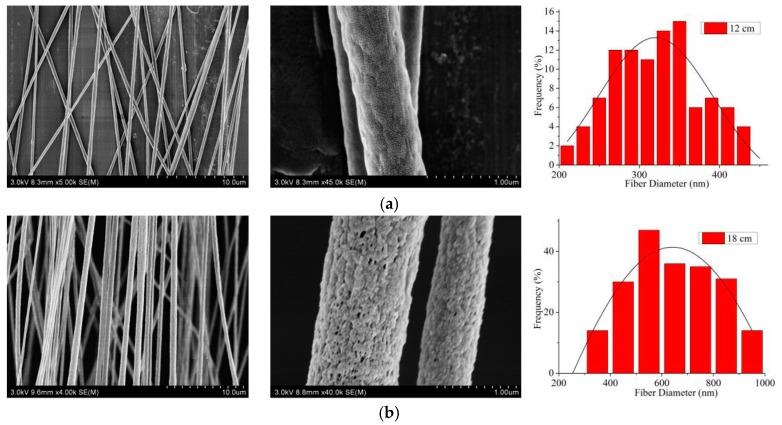
SEM pictures and the according diameter distribution (rightmost) of aligned porous CNFs with different electrospun tip-to-collection distances by MPEM (Average diameter: (**a**) 12cm, 321.1 ± 10.6 nm; (**b**) 18 cm, 579.9 ± 26.6 nm; (**c**) 24 cm, 252.0 ± 9.8 nm).

**Figure 10 nanomaterials-09-01782-f010:**
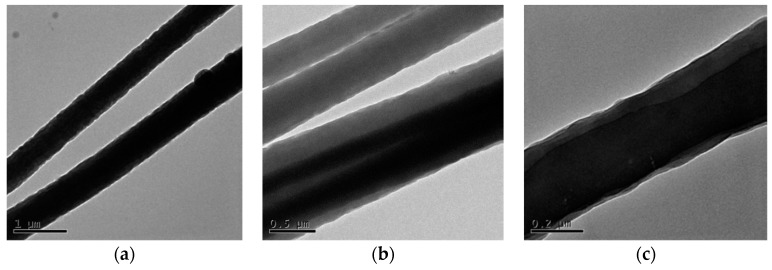
TEM pictures of aligned CNFs with nanopores by MPEM. (**a**) Gr agglomerates on the surface of the fiber; (**b**,**c**) is arranged along the axis inside the fiber.

**Figure 11 nanomaterials-09-01782-f011:**
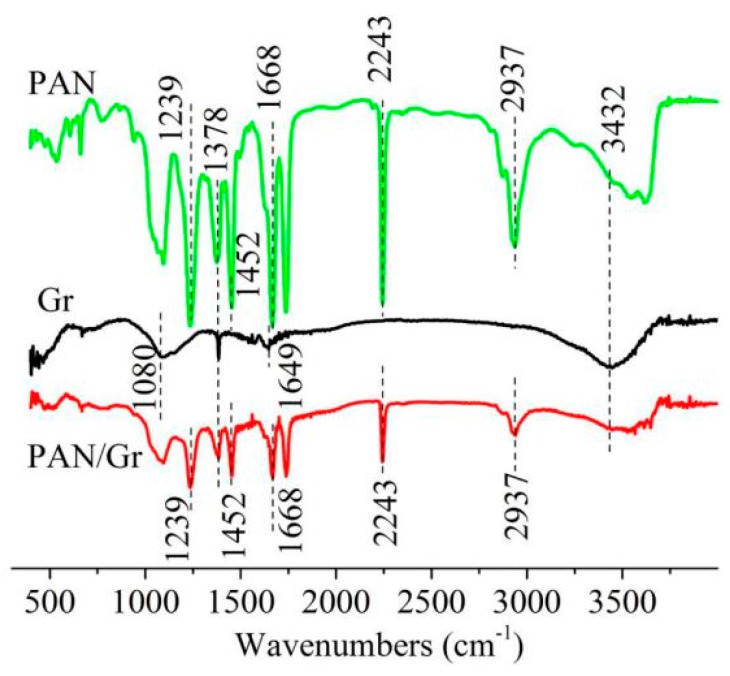
FTIR spectra of PAN nanofibers, Gr and aligned PAN/Gr CNFs with nanopores by MPEM.

**Figure 12 nanomaterials-09-01782-f012:**
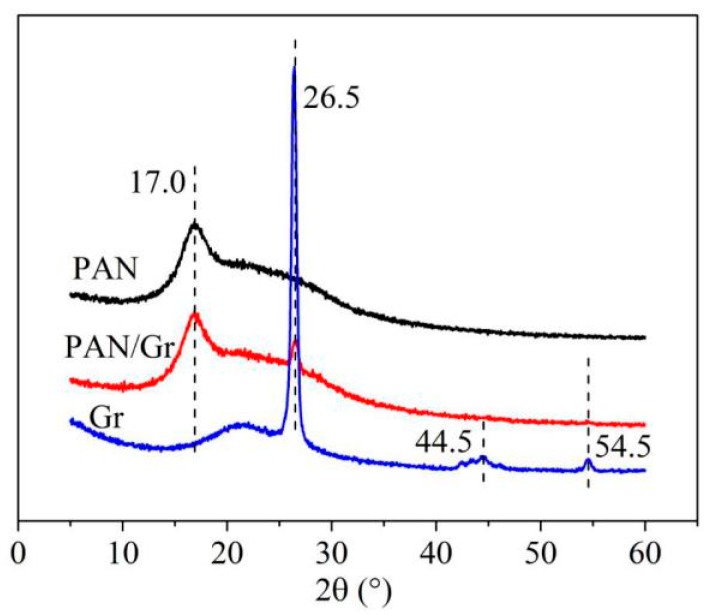
XRD results of PAN nanofibers, Gr and aligned CNFs with nanopores by MPEM.

**Figure 13 nanomaterials-09-01782-f013:**
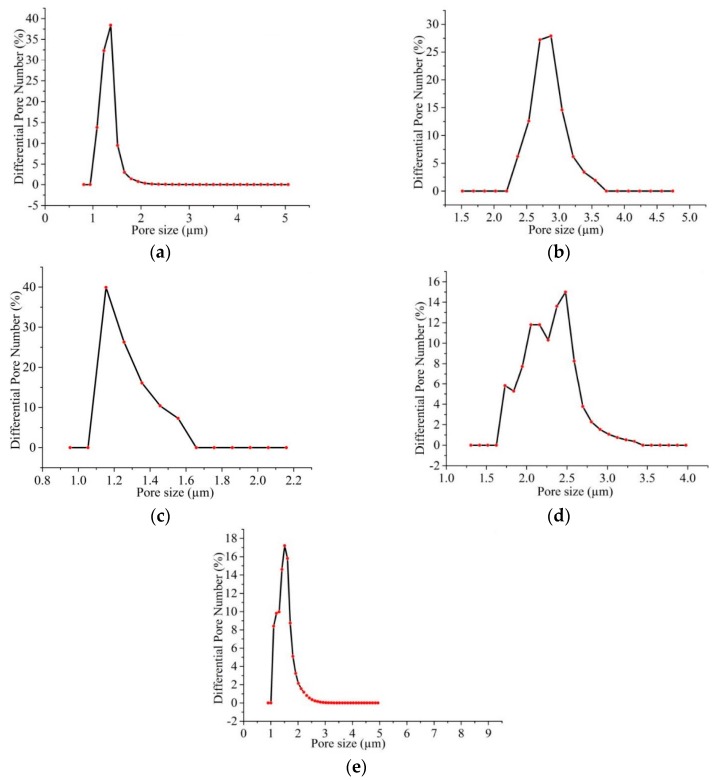
Pore size distributions of CNF membranes: (**a**) porous CNFs by MPEM; (**b**) porous CNFs by ES; (**c**) nonporous CNFs by MPEM; (**d**) nonporous CNFs by ES; (e) porous PAN NFs by MPEM.

**Table 1 nanomaterials-09-01782-t001:** The pore size distributions of CNF membranes.

Method	Membrane Consisted of Nanofibers	Pore Size (μm)	Maximum of Pore Number (Corresponding Pore Diameter)	Total Pore Number (/cm^2^)
MPEM	PAN nanofiber with nanopores	1.17–5.14	7.597 × 10^7^ (1.115 μm)	4.422 × 10^8^
	PAN/Gr CNF with nanopores	1.09–4.29	4.163 × 10^8^ (1.397 μm)	8.610 × 10^8^
	PAN/Gr CNF without nanopores	1.31–1.59	1.239 × 10^8^ (1.205 μm)	3.108 × 10^8^
Electrospinning (ES)	PAN/Gr CNF with nanopores	2.36–3.61	2.312 × 10^7^ (2.958 μm)	8.297 × 10^7^
	PAN/Gr CNF without nanopores	1.84–3.42	2.396 × 10^7^ (2.534 μm)	1.595 × 10^8^

**Table 2 nanomaterials-09-01782-t002:** The contact angle results of PAN and PAN/Gr CNF membranes with and without nanopores prepared by ES and MPEM.

Membrane Consisted of Nanofibers	ES	MPEM
PAN nanofiber with nanopores	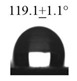	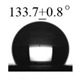
PAN/Gr CNF with nanopores	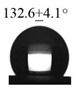	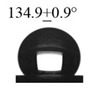
PAN/Gr CNF without nanopores	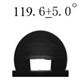	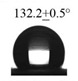

**Table 3 nanomaterials-09-01782-t003:** The surface resistances of different PAN/Gr CNF membranes.

PAN/Gr CNF Membrane	Surface Resistance (Ω)	
	MPEM	ES
CNF with nanopores	3.5 × 10^13^	4.2 × 10^13^
CNF without nanopores	1.0 × 10^11^	4.5 × 10^12^

**Table 4 nanomaterials-09-01782-t004:** The charge-transfer resistances of different carbonized CNFs.

Carbonizd CNFs	Carbonizd CNFs with Nanopores by MPEM	Carbonizd CNFs without Nanopores by MPEM	Carbonizd CNFs Without Nanopores by ES
Charge-transfer resistance (Ω)	0.3	0.6	1.0

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
