# Peer review of "Fabrication and Characterization of Electrospun Aligned Porous PAN/Graphene Composite Nanofibers"

_nanomaterials, 2019, doi:10.3390/nano9121782_

Round 1
Reviewer 1 Report
The authors have prepared porous aligned nanofibers using a parallel electrode electrospinning process, and have showed how including graphene nanoparticles can be introduced into the fibers. I have a few suggestions that I hope the authors will agree can improve the overall quality of the paper, which deals with a very interesting topic.
1. The grammar and general writing style require a careful revision as the intentions of the authors can sometime be misinterpreted. The word "electrospun" in the title also seem to be in subscript which suggests that the paper was not carefully reviewed for style (there are several other examples but I will refrain from listing them all).
2. The introduction can be improved. While it is understood what the goals of the research are, the authors make a series of very vague and general claims regarding the requirements for development of aligned porous nanofibers. I would expect to understand what exactly makes aligned nanofibers have such interesting properties, and why the porosity is important. Also, the authors should detail why the MPEM is a requirement or an advantage to produce these fibers as it is not made clear.
3. It is not clear how the alignment degree of the fibers was measured.
4. Figure 10 needs to be improved. The authors claim they prove that Gr were introduced into the CNFs aligned along the axis, but it is not clear what indicates this in the images.
5. The authors mention membranes throughout the text but it is not made clear anywhere where these are coming from.
6. The pore sizes reported on section 3.6 are very confusing. The pores are all >1um but the fibers all have diameters in the nm range. Is the porosity the measure of the pores on the membrane formed by the fibers? This is not made clear at all, as the authors constantly mention porous nanofibers so I was expecting to find very small pores on the surface of the actual fibers. This needs to be made clear as these are two very different concepts. If the authors in section 3.2.1 refer to porosity on the nanofibers surface they need to present some data supporting their statements as it is not clear how these pores evolve with different spinning conditions.
7. In the introduction the authors say "MPED method could make hydrophilic materials more hydrophilic" but there is no evidence of this at all in the contact angle results presented later on.
Reviewer 2 Report
In this research paper, a modified parallel electrode method (MPEM), was presented to fabricate aligned porous polyacrylonitrile/graphene (PAN/Gr) composite nanofibers (CNFs) in electrospinning progress. The effect of spinning parameters on fiber properties has been investigated and the optimum parameters have been determined. The authors claimed that the alignment degree and conductive properties of electrospun aligned porous CNFs were improved by MPEM which suggesting their potential application in electrochemical areas and electron devices.
- A critical point to be considered for publication of research is the novelty of the work. The purpose and novelty of the current research study have not defined clearly. The authors should provide a short review of the literature pertaining to the research topic. What are the advantages and novelty of their works in comparison to the following papers?
1. Hou J, Yun J, Byun H (Fabrication and Characterization of Modified Graphene Oxide/PAN Hybrid Nanofiber Membrane. Membranes 9:122.2019). 2. Lee J, Yoon J, Kim JH, Lee T, Byun H (Electrospun PAN–GO composite nanofibers as water purification membranes. Journal of Applied Polymer Science 135:45858.2018). 3. Mehrpouya F, Foroughi J, Naficy S, Razal J, Naebe M (Nanostructured electrospun hybrid graphene/polyacrylonitrile yarns. Nanomaterials 7:293.2017). 4. Song Y, Xu L (Permeability, thermal and wetting properties of aligned composite nanofiber membranes containing carbon nanotubes. International Journal of Hydrogen Energy 42:19961-19966.2017).
- Response surface methodology (RSM) is a collection of mathematical and statistical techniques for empirical model building. By careful design of experiments, the objective is to optimize a response (output variable) which is influenced by several independent variables (input variables). The central composite design (CCD) is one of the designs in response surface methodology (RSM) that can be very useful in the optimization process since it estimates the major effects and interactions that can be used to predict an optimum combination of factors by suggested model. To consider the major effects and also interactions (that is missing in current research) for optimizing the electrospinning conditions, I would suggest the authors use designed experiments such as RSM.
- The discussion on presented results is actually very limited. The brief descriptions of the presented results are confusing and ambiguous.
1. Introduction - The introduction requires a short review of the literature pertaining to the research topic. One approach may be to start with one or two paragraphs that introduce the reader to the general field of study. The subsequent paragraphs then describe how an aspect of this field could be improved. The final paragraph clearly states, what experimental question will be answered by the present study. The hypothesis is then stated. Next, briefly describe the approach that was taken to test the hypothesis. Finally, a summary sentence may be added stating how the answer of your question will contribute to the overall field of study.
- I would suggest the authors edit the introduction section according to the above-mentioned paragraph.
- I would suggest the authors remove the detailedobtained results from the introduction parts.
- The authors mentioned that:
“However, the electrospun porous nanofibers usually show randomly oriented structures and low alignment, which could result in weak conductivity and low molecular orientation to limit their practical uses”
The authors should provide a reference for this part.
- What are the advantages of “A modified parallel electrode method (MPEM),” in comparison to common rotating collector? How positively charged ring between the needle and the paralleled electrode collector can improve fiber formation and alignment?
- What is the amount of humidity during the spinning? 45+5%, or 50+5%? - I would suggest the authors replace Figure 1 from the introduction section to the method section. - The captions of all figures should remove from the text
2.3 Fabrication of Aligned porous PAN/Gr CNFs
- the type of a syringe and needle are missing
3.1 Property characterizations of spinning solutions
- The authors have investigated the effect of PAN and Gr concentration on viscosity and conductivity of solution in Figure 2 (a, b) but the concentration of water as solvent is missing. The authors should indicate what is the concentration of water for these results? (0 wt. %, 2 wt. %, 5 wt. % and 8 wt. %) - The authors should discuss how increasing the amount of Gr led to increasing the viscosity?
3.2.1 Effects of the concentrations of solute and solvent on the electrospun CNFs
- Why the effect of water concentration on fiber morphology has been investigated only in polymer concentration of 12 wt% and Gr concentration of 0.5 wt%? - - The following paragraph is not clear for me:
“It could be seen that there appeared visible pores on the CNFs surface after the addition of H2O due to the inconsistent volatilization time of H2O and DMF”.
- The effect of humidity on the formation of surface porosity has not been investigated. - Figure 4: What is the concentration of Gr? 0.5 wt%? The Gr concentration should be added to the figure caption. - The following paragraph is not clear for me, it is better the authors provide a reference for this method of the alignment measurement and the method should be described in method section
“The degree of nanofiber alignment was defined as the ratio of the number of nanofibers, whose θ is between -9° and 9°, to the total number of nanofibers”
3.4 FTIR spectra analysis
- The authors mentioned that:
“In the Gr spectra, the peak at 1378 cm-1 corresponded to C-O-C, thus certificated that Gr wasn’t restored completely”
- Is there Oxygen in the chemical structure of graphene nanoplates??
3.5 XRD spectra
- The authors should provide the reference for picks related to the graphene.
3.6 Porosity, Wetting, and Conductivity Properties
- The authors should clearly describe what is the difference between “Porous Gr/PAN” and “Nonporous Gr/PAN” in Table 1.
- The following part is not clear for me:
“It was seen that with the addition of Gr and the appearance of a porous structure on the surface of the CNFs, the pore size of the CNF membrane decreased and the according pore number increased. At the same time, the MPEM method endowed the aligned CNF membrane smaller pore size but more pore number”.
- In this section authors results that:
“Therefore, the MPEM method could provide hydrophobic materials more hydrophobic”
While in the introduction section they mentioned that:
“In addition, the MPEM method could make hydrophilic materials more hydrophilic, and the alignment of the electrospun CNFs could improve the electrical conductivity”.
Finally it is not clear that if this method has improved the hydrophilicity and hydrophobicity!
Grammar
English usage in this manuscript must be substantially improved. There are many grammatical errors and vague descriptions. I would suggest authors to ask a native English to edit the manuscript.
The spinning parameters, such as the concentrations of solute and solvent, spinning voltage, spinning distance, and et al., were discussed in this study, and the optimum parameters were determined.
It is expected to generate aligned porous nanofibers to broaden the applications, such as electrochemical devices, optoelectronic sensors, reinforcements, and et al.
In addition, it is generally known that single or pure??polymer nanofibers can’t satisfy the need for electrochemical devices.
Round 2
Reviewer 1 Report
Thank you for attending to my suggestions.
Author Response
Thank you for your review!
Reviewer 2 Report
The authors followed 60%-70% of our indications. Therefore, even if the mauscript improved, it is not yet possible to accept it in its current form. The following points must be fulfilled.
1. The authors indicated that the novelty of their work is based on the production of aligned composite nanofibers (CNFs) with surface porosity via a modified parallel electrode method. They believe that these pores not only increase the specific surface area of the CNFs but also improve the transport and transfer efficiency of ions when they are used as the electrode materials. But according to their final results, the presence of surface porosity led to an increase in surface resistance resulting in reduced conductivity! According to these results, what are the advantages of surface porosity as one of the novelties of their work?
2. The authors responded that “There is Oxygen in the chemical structure of Gr nanoplates. This may be because the graphene materials used in this experiment weren’t restored completely, resulting in the presence of Oxygen in the Gr nanoplates”. For me, this is not an acceptable reason.
3. The author responded that “We have revised and added the discussion on presented results and make it clear”. But actually there is no significant difference in results and discussion part.
4. The provided references (49, 50) are related to the XRD of graphene oxide, not grapheme!
5. The magnification of SEM images of figure 4 and 6 are not the same.
6. There is no scale bar in SEM images of figure 9!
7. The authors should indicate all electrospinning conditions including the spinning voltage, the ring voltage, the distance of two paralleled electrodes and the spinning distance for each series of their experiments. (Figure 3 to 9).
8. The captions of all figures have still remained in the text.
9. The conditions of electrospinning (ES) for the production of random fiber is missing.
Round 3
Reviewer 2 Report
The revised manuscript is acceptable.